# Teaching Large Language Models to Reason with Reinforcement Learning

**Alex Havrilla** [1 2]   **Sharath Raparthy** [1]   **Christoforos Nalmpantis** [1]   **Jane Dwivedi-Yu** [1]   **Maksym Zhuravynski** [3]
**Eric Hambro** [4]   **Roberta Raileanu** [1]

## Abstract

Reinforcement Learning from Human Feedback (**RLHF**) has emerged as a dominant approach for aligning LLM outputs with human preferences. Inspired by the success of RLHF, we study the performance of multiple algorithms that learn from feedback (Expert Iteration, Proximal Policy Optimization (**PPO**), Return-Conditioned RL) on improving LLM reasoning capabilities. We investigate both sparse and dense rewards provided to the LLM both heuristically and via a learned reward model. We additionally start from multiple initializations with and without supervised fine-tuning (**SFT**) data. Overall, we find models fine-tuned with Expert Iteration to consistently achieve the highest task accuracy with PPO and RCRL close behind. Surprisingly, the sample complexity of Expert Iteration is similar to that of PPO, requiring at most on the order of $10^6$ samples to converge from a pretrained checkpoint. We investigate why this is the case, concluding that during RL training models fail to explore significantly beyond solutions already produced by SFT models. Additionally, we discuss a trade off between maj@1 and pass@96 metric performance during SFT training and how conversely RL training improves both simultaneously. We then conclude by discussing the implications of our findings for RLHF and the future role of RL in LLM fine-tuning.

## 1. Introduction

The reasoning abilities of large language models (**LLMs**) are rapidly improving as measured by their performance on numerous math, science and code benchmarks (Cobbe et al., 2021; Hendrycks et al., 2021b; Sawada et al., 2023; Liang et al., 2022; Srivastava et al., 2022; Rein et al., 2023; Mialon et al., 2023; Chollet, 2019; Mishra et al., 2022; Hendrycks et al., 2021a; Austin et al., 2021; Patel et al., 2021; Gao et al., 2021). Simultaneously, Reinforcement Learning from Human Feedback (RLHF) (Bai et al., 2022; Ziegler et al., 2019; Ouyang et al., 2022) and instruction fine-tuning (Wei et al., 2021; Mishra et al., 2021) have made significant progress in aligning LLMs with human preferences. Improvements in model instructability have further increased apparent model capability by making complex behaviors more accessible via instruction prompting. This has led to a number of increasingly sophisticated prompting strategies augmenting LLM reasoning capabilities such as Chain-of-Thought (Wei et al., 2022) or Tree-of-Thoughts (Yao et al., 2023).

Previous work in reinforcement learning (RL) such as AlphaGo (Silver et al., 2017), AlphaStar (Vinyals et al., 2019), and OpenAI Dota 2 (Berner et al., 2019) demonstrate that RL techniques can be used to train neural networks capable of sophisticated planning and reasoning in game environments. Cicero (Bakhtin et al., 2022) in particular succeeds in combining an RL trained planning agent with a dialogue fine-tuned LLM to achieve nearly super-human performance in the board game Diplomacy. Given these previous successes and the inherent interactive nature of problem solving, applying RL to LLM reasoning seems a natural next step. In this paper, we study how ideas from RL can be used to improve the reasoning capabilities of LLMs across a variety of reward schemes and model initializations.

We begin by comparing the performance of different RL algorithms on reasoning tasks $\tau$ defined as a distribution of question answer tuples $(Q, A)$. The task $\tau$ can be extended to define a *Markov Decision Process* (**MDP**) 4-tuple $(\mathcal{S}, \mathcal{A}, P_a, R_a)$ where tokens serve as both actions and accumulated state with deterministic dynamics. By default we use a sparse reward of $+1$ if the final answer is correct but also experiment with dense rewards matching intermediate steps in a reference solution and rewards synthetically generated using a reward model. We evaluate models with 7B and 13B parameters both starting from supervised fine-tuned (SFT) checkpoints and pre-trained checkpoints. We report four metrics assessing model performance on a task specific test set: 1) maj@1 score computed by greedily sampling once per question, 2) maj@96 score computed by

---

[*]Equal contribution   [1]Meta [2]Georgia Tech [3]StabilityAI [4]Anthropic.   Correspondence to:   Alex Havrilla <ahavrilla3@gatech.edu>.

*The first AI for MATH Workshop at the 41st International Conference on Machine Learning, Vienna, Austria. Copyright 2024 by the author(s)*

sampling K = 96 times per question and uniformly voting on the final answer, 3) rerank@96 score computed by sampling K = 96 times and choosing the final answer using an Outcome-Based Reward Model (**ORM**), and 4) pass@96 score computed by sampling the model K = 96 times and taking the best result according to the ground truth answer.

We find that overall the simplest method, Expert Iteration (**EI**) (Anthony et al., 2017), performs best across all metrics for most reward setups and model initializations. Even more surprisingly, EI is nearly as sample efficient as more sophisticated algorithms like Proximal Policy Optimization (**PPO**), both requiring only a few thousand samples to converge. We also observe the gap between pretrained model performance and SFT model performance shrinks ($< 10\%$ gap on GSM8K) after RL fine-tuning, with larger models having a smaller gap. Additionally, previous work identified a trade-off between test time maj@1 performance and pass@96 performance during supervised fine-tuning (Cobbe et al., 2021), with continued training increasing maj@1 score at the expense of pass@96 score. We identify the limited diversity of the dataset as a core reason for this. We show that RL fine-tuning can improve both metrics simultaneously due to the fact that RL generates its own data during training, resulting in a more diverse set of examples to learn from.

We then discuss why EI and return conditioned RL are competitive with PPO, suggesting two principal factors. Firstly, the reasoning tasks we consider have entirely deterministic dynamics: a setting in which direct behavior cloning and return conditioned RL is known to do well (Brandfonbrener et al., 2022). In contrast, PPO often succeeds in environments with a high degree of stochasticity (Bhargava et al., 2023). Second, we identify a lack of sophisticated exploration carried out by models during RL fine-tuning. This limitation significantly impacts any performance or sample complexity advantages PPO may have when fine-tuning the pretrained model. We come to this conclusion from a number of observations, noting in particular quickly saturating pass@96 scores early in RL training. We conclude with a discussion of the impacts of our observations on RLHF and the future of LLM fine-tuning via RL.

In summary we make the following contributions:

- A comprehensive study of PPO fine-tuning of LLMs on reasoning tasks using different types of rewards, model sizes and initializations.

- A comparison to expert iteration and return-conditioned RL from which we find expert iteration reliably attains the best performance and competitive sample complexity across the board.

- A discussion of the implications of our findings for RLHF and the future of RL fine-tuning for LLMs, iden-

tifying exploration as a major limiting factor.

## 2. Related Work

**LLM Reasoning:** Recent work combines base LLM reasoning capabilities with planning and search algorithms to further boost performance on a wide range of tasks (Yao et al., 2023; Besta et al., 2023; Ye et al., 2022; Yao et al., 2022; Dohan et al., 2022). Tree of thought (Yao et al., 2023) for example combines LLMs with a breadth first search algorithm, relying on the LLM to both propose actions and evaluate state. Other works combine LLMs with tools (Schick et al., 2023; Qin et al., 2023; Zhou et al., 2023a) further boosting reasoning capability. Combining GPT-4 with a python code interpreter for generation and self-verification achieves an impressive 84% on the hard MATH benchmark (Hendrycks et al., 2021a; Zhou et al., 2023a).

Other works focus on LLMs for mathematical reasoning in natural language (Cobbe et al., 2021; Lewkowycz et al., 2022; Azerbayev et al., 2023; Lightman et al., 2023; Patel et al., 2021; Zhu et al., 2023; Rafailov et al., 2023). Particularly relevant to our study is Cobbe et al. (2021) which fine-tunes GPT-3 on supervised math word problem (**MWP**) reasoning traces. In addition they train solution verifiers called Outcome Based Reward Models (**ORMs**) which predict the probability of correctly solving a question $Q$ giving a prefix of intermediate steps $P_i = (S_1, ..., S_i)$ i.e. $p(is\_correct(A)|Q, P_i)$ where $A$ is a solution with prefix $P_i$. Process based reward models (**PRMs**) (Uesato et al., 2022; Lightman et al., 2023) can also be trained to instead look at the step-level accuracy of solutions. More recent work (Luo et al., 2023) utlizies a PRM distilled from GPT-4 feedback as a reward signal during PPO.

**RL for LLM fine-tuning:** Reinforcement Learning from Human Feedback (RLHF) is perhaps the most well-known application of RL techniques for fine-tuning LLMs. RLHF (Christiano et al., 2017; Ziegler et al., 2019; Stiennon et al., 2020; Ouyang et al., 2022; Bai et al., 2022; Glaese et al., 2022; Peng et al., 2021; Ramamurthy et al., 2022) most often works by training a *reward model* to capture human preferences over a task $\tau$. The reward model is then used to score LLM responses to prompts from the task after which policy improvement is performed. PPO is most often used (Ouyang et al., 2022; Bai et al., 2022) but several recent works including ReST (Gulcehre et al., 2023), Reward-Ranked Fine-tuning (Dong et al., 2023), and AlpacaFarm (Dubois et al., 2023) all demonstrate simply fine-tuning on high return responses with the standard cross-entropy loss can attain comparable performance. We broadly refer to this class of algorithms as Expert Iteration.

A large body of work studying RL for LLM fine-tuning also exists outside of the RLHF sphere. Work on text games (Yao et al., 2020; Ammanabrolu & Riedl, 2019) and other

interactive textual environments (Zhou et al., 2023b; Carta et al., 2023) seek to ground LLMs via interaction and RL. RL has also been applied to improving model performance on controllable generation and question answering tasks (Lu et al., 2022; Liu et al., 2022). Various forms of expert iteration have also been applied to improve LLM reasoning capabilities (Huang et al., 2022; Yuan et al., 2023; Zelikman et al., 2022; Uesato et al., 2022). For example "Scaling Relationship on Learning Mathematical Reasoning with Large Language Models" (Yuan et al., 2023) applies a single round of expert iteration across multiple model sizes on GSM8K. They observe sizeable gains in all metrics for smaller models, with gains diminishing for larger models. A related body of work studies RL for code generation (Le et al., 2022; Shen et al., 2023; Rozière et al., 2023). Shen et al. (2023) in particular reports a huge increase in StarCoder's (Li et al., 2023) maj@1 performance after a single round of expert iteration, jumping from ~30% to ~60%.

Despite all the above work, it remains unclear exactly what factors account for the biggest impact during RL fine-tuning due to wide variance in tasks, pre-training data, supervised fine-tuning data, RL algorithm used, and the reward source. Our work conducts a thorough analysis of all these factors to understand exactly how different algorithms compare when applied to improving LLM reasoning capability. As a result we are able to identify key bottlenecks to further LLM improvement via RL and provide a discussion on promising future directions.

## 3. Methods

**Reasoning as an RL problem** We study the performance and sample complexity requirements for various RL algorithms when fine-tuning LLMs on reasoning tasks. We consider Expert Iteration (EI) (Anthony et al., 2017), Proximal Policy Optimization (PPO) (Schulman et al., 2017), and Return-Conditioned RL (RCRL) (Brandfonbrener et al., 2022) as representative algorithms from the RL literature. In general, the goal of all RL algorithms is to maximize the expected future return $\mathbb{E}_{A \sim \pi(Q), (Q, \cdot) \in \tau} R(A)$ of a student policy $\pi$ on task $\tau$. We call the highest return policy the *optimal policy* $\pi^*$. Each of our chosen algorithms goes about finding $\pi^*$ in a different way.

**PPO** is an example of an *online* RL algorithm. Online algorithms engage in both an exploration phase and a policy improvement phase which updates $\pi_\theta$ using data generated during the exploration phase. PPO is also an *on-policy* algorithm which samples model rollouts during exploration from the student policy $\pi_\theta$ being trained. During policy improvement, the student $\pi_\theta$ updates its parameters via gradient descent by directly maximizing for reward with the objective

$$J(\theta) = \mathbb{E}_t \left[ min(\frac{\pi(a_t|s_t)}{\pi_{\text{old}}(a_t|s_t)} \hat{A}_t, clip(1 - \epsilon, 1 + \epsilon, \frac{\pi(a_t|s_t)}{\pi_{\text{old}}(a_t|s_t)}) \hat{A}_t) \right]$$

where $\hat{A}_t$ estimates the *advantage* i.e. difference between $Q(s, a)$ (the expected return after taking action $a$ at state $s$) and value $V(s)$ (the expected return at state $s$).

In practice, for PPO we sample 1024 rollouts at a time with a temperature of 0.7 and $N = 4$ rollouts per question. Training is then run on these samples for $K = 4$ PPO epochs with a batch size of 32. Additionally, we train using LoRA (Hu et al., 2021) with $r = 128$. Training is run for 4000 gradient steps. The best checkpoint is then selected via performance on a validation set.

**Expert iteration** is also online but more off-policy than PPO. An initial expert policy approximation $\hat{\pi}_0^*$ is sampled on the entire train set $K$ times per question before any policy improvement. The $\hat{\pi}_0^*$ is often constructed using repeated sampling from an initial policy $\pi_0$. For example, AlphaZero (Silver et al., 2017) and subsequent work (Schick et al., 2023) combine $\pi_0$ with Monte Carlo Tree Search. Sampling $\hat{\pi}_0^*$ constructs an initial set of rollouts $D_1$ which are then distilled back into a policy $\pi_1$ via a standard cross-entropy loss: $\sum_{\tau \in D} \sum_{t=1}^{H} -log(\pi_\theta(a_t|s_t))$. This process can be repeated to construct policy $\pi_i$ fine-tuned on dataset $D_i = R_i \cup D_{i-1}$ where $R_i$ corresponds to exploration done by $\pi_{i-1}$.

In our setting we construct an approximation to the optimal policy $\hat{\pi}^*$ by rejection sampling our student policy $\pi_\theta$. After generating $K$ samples $S_1, ..., S_K$ on a question $Q$ we construct $D_1$ by filtering all $(Q, S_i)$ pairs with return below a threshold $T$. De-duplication is then performed on the remaining samples.

In practice, during the expert iteration exploration phase we sample each question in the train set $K = 96$ times with temperature $T = 1.0$. To construct the training set we filter out incorrect solutions and duplicates. Importantly, fine-tuning is then done from the pretrained base model with the same hyperparameters as SFT. This is repeated until performance on a validation set saturates.

**Return Conditioned RL** Return conditioned RL algorithms seek to train policies conditioned on both the current state $s$ and desired return $R$ when sampling an action. This is motivated by a desire to learn return conditionable policies which can change depending on the desired return. Best performance can then be sampled by conditioning on the highest possible return.

We consider an offline version of this class of algorithms similar to a decision transformer (Chen et al., 2021). A training dataset $D$ is constructed by generating state, action, return $\tau = ((s_t, a_t, g_t))_{t=1}^{H}$ trajectories. Training

is done by predicting the action given state and return: $\sum_{\tau \in D} \sum_{t=1}^{H} -log(\pi_\theta(a_t|s_t, g_t))$. In practice we construct $D$ by sampling solutions $S = (S_1, ..., S_L)$, where each $S_i$ is an intermediate step, from our best EI trained policy $\pi_{\text{EI}}$ given a question $Q$. We generate return labels for each step $S_i$ by sampling $\pi_{\text{EI}}$ K many times from $P_i = (S_1, ..., S_i)$. This results in binary labels $l_1, .., l_K$ evaluating the correctness of the generated final answers. $S_i$ is then labeled as "[GOOD]" if the average return $\frac{1}{K}\sum_{k=1}^{K} l_k \geq T$ and otherwise is labeled as "[BAD]". Typically we set $T = 0.5$. We then filter the dataset to ensure a balanced number of correct and incorrect solutions. See Section F in the appendix for more details about the step-label generating process.

**Outcome Based Reward Modeling** Multiple works (Cobbe et al., 2021; Uesato et al., 2022) train Outcome Based Reward models **ORMs** as *verifiers* of candidate solutions to word problems. The ORM can then be used to rerank multiple candidate solutions generated by a student model, significantly boosting performance. Training data for the ORM is generated using a student policy $\pi$ by sampling $K$ solutions per question $Q$ in the task dataset. The ORM is trained as a classifier by predicting the probability of reaching the correct final answer $p(\text{is\_correct(A)}|Q, P_i)$ from an intermediate sequence of steps $P_i = (S_1, ..., S_i)$, $P_i \subseteq A = (S_1, ..., S_L)$.

# 4. Experiments

We conduct our evaluations on GSM8K and SVAMP (Patel et al., 2021): two math word problem benchmarks. In addition on GSM8K we consider two data regimes: first with SFT data and then without SFT data. We evaluate all models using greedy sampling (maj@1) accuracy as well majority vote at 96 samples (maj@96), ORM based reranking at 96 samples (rerank@96), and best of 96 sample (pass@96) accuracy. Unless otherwise specified, test-time sampling is done greedily for maj@1 and with a temperature of 0.7 otherwise. We sample the RCRL models one step/line at a time, conditioning on the "[GOOD]" token. We note while the notion of a "step" is not clearly defined in general, in our case we can simply regard each step as ending with a sentence or newline. All experiments are done using instruction-tuned Llama 2-chat 7B and Llama 2-chat 13B models.

## 4.1. Results with SFT Initialization

When given access to SFT data, we first supervise fine-tune Llama 2-chat models for 4 epochs with a global batch size of 128 and an initial lr of 2e-5 decayed to 2e-7 with a cosine warmup schedule. We call the resulting models **SFT**. When fine-tuning with PPO we initialize using this checkpoint. In contrast, for both EI and RCRL we generate data with the SFT checkpoint but reset training to start from the pretrained

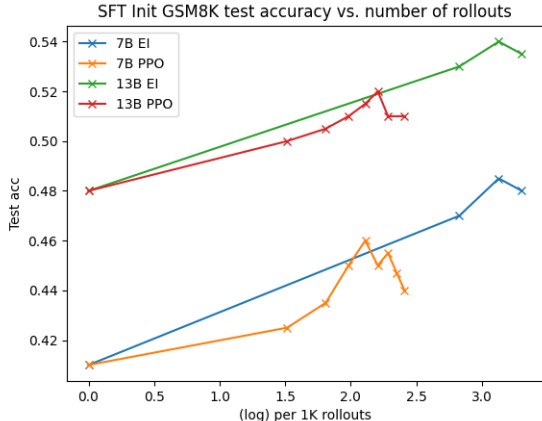

*Figure 1.* Sample complexities of SFT initialized models on GSM8K. EI achieves better performance than PPO with the same order of magnitude of samples.

base model. Similarly to Zelikman et al. (2022), we find this model resetting is crucial for achieving best performance. Results for both 7B and 13B models are reported in Table 1.

**Expert iteration achieves the best performance with competitive sample complexity** Surprisingly, we find EI achieves the best performance with a maj@1 accuracy of 0.485 and 0.53 on 7B and 13B models respectively. For both model sizes the best greedy accuracy is achieved after $n = 2$ expert iterations, after which performance plateaus. In total, EI gives a sizable improvement of around 7% over the SFT baseline. Similar gains can be seen in maj@96, rerank@96, and pass@96 scores with.

PPO models underperform EI, with ORM guided PPO giving the biggest improvement of around 5% over the SFT baseline. Again, maj@96, rerank@96, and pass@96 accuracies show similar improvements. Interestingly, despite further training on top of the SFT initialization, PPO models retain competitive rerank@96 and pass@96 scores when compared to regression we see after further supervised fine-tuning. We believe this is due to the relatively more diverse nature of the exploration dataset used to update the model.

Finally, RCRL models under-perform EI models despite training on EI generated data with an even balance between '[GOOD]' and '[BAD]' step labels. This matches similar results from (Du et al., 2023) which use only sparse labels for the entire rollout. Further, when sampling the RCRL model unconditionally the model often generates the perfectly valid steps following a '[BAD]' label resulting in a correct final answer. These results suggest RCRL models are not correctly learning what constitutes a '[GOOD]' versus '[BAD]'. This suggests RCRL models are unable to usefully incorporate information from partially correct solutions at train time. An ablation (See sec. A of the ap-

| | maj@1 | | maj@96 | | rerank@96[†] | | pass@96 | |
|---|---|---|---|---|---|---|---|---|
| | 7B | 13B | 7B | 13B | 7B | 13B | 7B | 13B |
| SFT | 0.41 | 0.48 | 0.47 | 0.53 | 0.54 | 0.68 | 0.72 | 0.84 |
| $EI_n$ | **0.48** | **0.53** | **0.55** | **0.59** | 0.64 | **0.71** | 0.8 | **0.88** |
| ORM $EI_n$ | **0.48** | **0.53** | 0.54 | 0.58 | **0.65** | **0.71** | **0.81** | 0.87 |
| ORM RCRL | 0.45 | 0.51 | 0.5 | 0.56 | 0.54 | 0.69 | 0.73 | 0.83 |
| Sparse PPO | 0.44 | 0.51 | 0.49 | 0.55 | 0.58 | 0.67 | 0.77 | 0.85 |
| Dense PPO | 0.43 | 0.50 | 0.47 | 0.54 | 0.53 | 0.65 | 0.71 | 0.81 |
| Sparse ORM PPO | 0.46 | 0.51 | 0.51 | 0.55 | 0.59 | 0.67 | 0.79 | 0.83 |
| Dense ORM PPO | 0.46 | 0.51 | 0.52 | 0.55 | 0.59 | 0.67 | 0.76 | 0.83 |
| Llema[*] | 0.40 | 0.62 | 0.54 | 0.69 | N/A | | N/A | |
| RFT | 0.47 | 0.54 | 0.58 | 0.65 | N/A | | N/A | |
| WizardMath | 0.55 | 0.64 | N/A | | N/A | | N/A | |
| GPT-3[**] | 0.2 | 0.31 | N/A | | 0.39 | 0.55 | 0.71 | NA |
| GPT-4[***] | 0.91 | | N/A | | N/A | | N/A | |

*Table 1.* Results when initializing from SFT. $EI_n$ denotes n rounds of expert iteration until convergence with $n = 2$ for 7B and $n = 2$ for 13B. [†]Note all reranking is done using an ORM trained with samples from $EI_n$. Results from other works are included on the bottom for reference. N/A stands for not available. [*]Llema results reported for 7B/34B sizes without fine-tuning. [**]GPT-3 results reported for 7B/175B sizes. [***]GPT-4 size unknown.

pendix) on the ratio of positive to negative labels finds a balanced ratio yields the worst performance, with increasing the amount of positive data leading to better results.

In Figure 1 we plot the number of model rollouts against model performance in log-scale. PPO models achieve their best accuracies after around 60,000 rollouts while EI models train with an order of magnitude more. However, the resulting train time in both cases is about a day. This is largely due to memory requirements from PPO, resulting in lower rollout throughput and smaller mini-batch sizes at train time. Additionally, in the SFT case we did not experiment with reducing the number of samples from $K = 96$ per question for EI. However, we expect this number can be significantly reduced without impacting performance. For a more thorough investigation of sample complexity requirements, see Figure 4.

**Extra guidance from ORMs or dense rewards provides little benefit** Overall, the ORM slightly improves PPO performance and negligibly impacts EI performance. For both algorithms it provides an improvement in terms of sample complexity. However, this does not change final performance. See Figures 2 and 3 which plot the performance against number of model rollouts for differnt reward regimes.

Giving dense rewards at best provides no extra benefit to performance when given either heuristically or via the ORM. Giving a heuristic dense reward even slightly harms model performance relative to the sparse setting. Recall we give intermediate reward by comparing intermediate model generated steps to the reference solution. This likely encourages more overfit to exact solutions in the train set, limiting solution diversity.

**RL improves maj@1 accuracy without impacting pass@96 performance** Looking at the pass@96 accuracies more closely, we see most similarly sized models are within 3% of the best result. This demonstrates with enough sampling, most models are able to solve a very similar range of problems. Further, while the pass@96 accuracy of our best EI model initially seems much higher than the SFT checkpoint, this is only because the SFT checkpoint has undergone much more training on a less diverse dataset. Simply supervised fine-tuning for half as many steps results in a checkpoint with maj@1 = 0.36 but pass@96 = 0.76. This further suggests RL training mostly impacts maj@1 accuracy without significantly improving on a pass@n accuracy which can be achieved with a light amount of supervised fine-tuning.

The proximity of pass@96 accuracies among most models is in sharp contrast to the rerank@96 performance. Here we find $EI$ models enjoy around a 5% lead over other models. At first glance this seems contradictory with relatively similar pass@96 performance. However, we believe a nontrivial percentage of this gap is due to **overfit of the ORM to the EI model which was used to generate its training data**.

### 4.2. Results with no SFT Initialization

We now consider the case when no SFT data is available for training. For questions in both SVAMP and GSM8K we give pretrained models access to a two-shot prompt with samples drawn from the GSM8K validation set. For EI models, we remove these prompts after the first round of exploration, instead relying on the generated SFT data. As in the case with SFT data, we run both algorithms until performance saturates. For PPO this happens after 250 steps

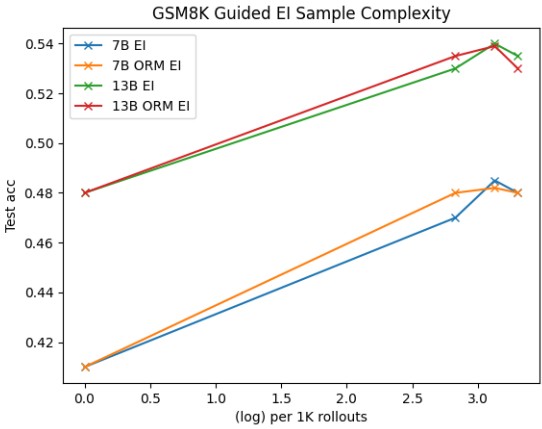

*Figure 2.* maj@1 scores of EI and ORM aided EI models over the course of training. The ORM improves sample efficiency but not performance.

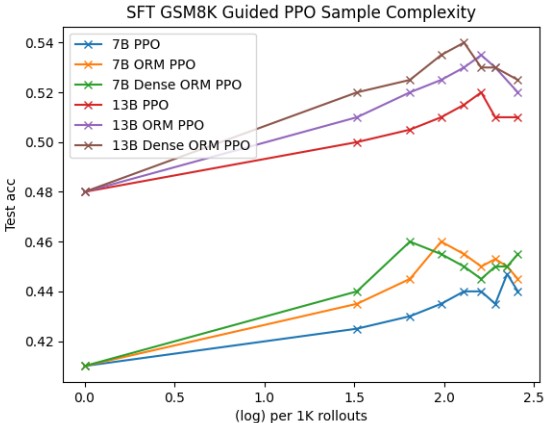

*Figure 3.* maj@1 scores of PPO and ORM guided PPO models over the course of training. As with EI models, the ORM improves sample efficiency but not performance.

on SVAMP and 1000 steps on GSM8K. For EI, this happens after $n = 5$ rounds of exploration and distillation. Results on both datasets are reported in Tables 2 and 3.

**EI achieves the best performance overall** Even without SFT data, EI achieves the best performance on SVAMP, improving 7B/13B pretrained greedy model accuracies over 50% from 0.06/0.05 to 0.58/0.69%, respectively. PPO performs slightly better than EI on GSM8K, improving from 0.05/0.03 to 0.31/0.4. Both algorithms achieve comparable pass@96 scores across modes sizes, further supporting our observations from the SFT regime that EI mostly improves maj@1 scores relative to PPO. The prompted 13B model on GSM8K even attains 0.83 pass@96 accuracy which is close to the 0.84 pass@96 score achieved by the SFT model, despite having no access to SFT data itself.

|  | maj@1 | | maj@n | | rerank@n$^\dagger$ | | pass@n | |
|---|---|---|---|---|---|---|---|---|
|  | 7B | 13B | 7B | 13B | 7B | 13B | 7B | 13B |
| Prompted | 0.05 | 0.03 | 0.14 | 0.18 | 0.17 | 0.24 | 0.22 | 0.27 |
| EI$_n$ | 0.31 | 0.4 | 0.35 | 0.47 | 0.39 | 0.63 | 0.45 | **0.83** |
| ORM EI | 0.28 | 0.37 | 0.33 | 0.43 | 0.37 | 0.59 | 0.42 | 0.76 |
| Sparse PPO | **0.32** | **0.41** | **0.37** | **0.48** | **0.41** | **0.65** | **0.5** | **0.83** |
| Sparse ORM PPO | 0.29 | 0.38 | 0.34 | 0.44 | 0.4 | 0.62 | 0.49 | 0.81 |
| Dense ORM PPO | 0.29 | 0.39 | 0.35 | 0.45 | 0.41 | 0.64 | 0.5 | 0.82 |

*Table 2.* Results for 7B/13B models when **not** using SFT initialization on GSM8K. Sparse PPO performs slightly better than EIin this setting. *Note all reranking is done using an ORM trained with samples from EI$_n$ model.

|  | maj@1 | | maj@n | | rerank@n$^\dagger$ | | pass@n | |
|---|---|---|---|---|---|---|---|---|
|  | 7B | 13B | 7B | 13B | 7B | 13B | 7B | 13B |
| Prompted | 0.06 | 0.05 | 0.2 | 0.25 | 0.24 | 0.29 | 0.3 | 0.36 |
| EI$_n$ | **0.58** | **0.69** | **0.6** | **0.75** | **0.62** | **0.78** | **0.70** | **0.93** |
| Sparse PPO | 0.44 | 0.51 | 0.55 | 0.66 | 0.58 | 0.73 | 0.72 | 0.89 |
| Sparse ORM PPO | 0.43 | 0.51 | 0.52 | 0.64 | 0.54 | 0.71 | 0.65 | 0.85 |
| Dense ORM PPO | 0.44 | 0.52 | 0.51 | 0.63 | 0.55 | 0.73 | 0.67 | 0.85 |

*Table 3.* Results for 7B/13B models when **not** using SFT initialization on SVAMP. EI$_n$ denotes the best EI model after $n$ iterations. EI outperforms PPO.

**EI has the same sample complexity as PPO** As before we plot the reward versus number of model rollouts for PPO and EI in Figures 4 and 5. On GSM8K PPO models attain their best maj@1 accuracies after only 30,000 rollouts and on SVAMP even less. Surprisingly, EI models have the same sample complexity as PPO on SVAMP, requiring more samples to converge but also converging to a much higher accuracy. EI still appears to have higher sample complexity on GSM8K, however as noted before this may be due to oversampling each prompt during the exploration phase. To test this, we reduce the number of samples per prompt each round of EI from $K = 96$ to $K = 4$. The resulting EI models require more iterations to converge but require far less total samples, also converging in accuracy only a few percentage points lower than $K = 96$ samples per prompt. With $K = 4$ rollouts per prompt **EI has the same sample complexity as PPO** on GSM8K.

This is a particularly surprising finding when compared to the performance of EI and PPO on more classical RL problems training a neural network from scratch. Often PPO enjoys far better sample complexity in these settings. One major difference here is the initialization of our student from a pretrained model which imparts a very strong bias on the kind of behaviors and exploration encountered during RL training. Both the extremely small sample complexity and the comparability of EI and PPO in this setting provide more evidence that models are not truly engaging in complex exploration, but instead primarily drawing on what they already know from the pre-training phase.

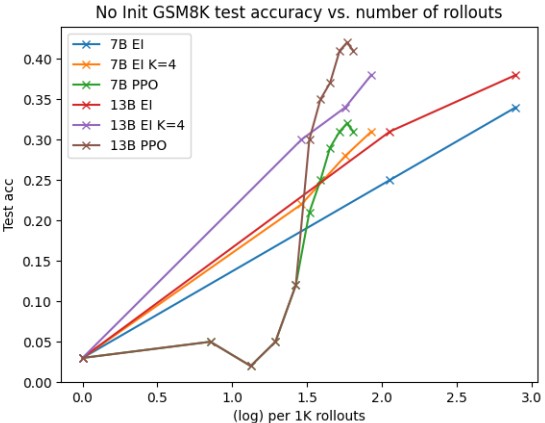

*Figure 4.* Sample complexities on GSM8K from pretrained initialization.

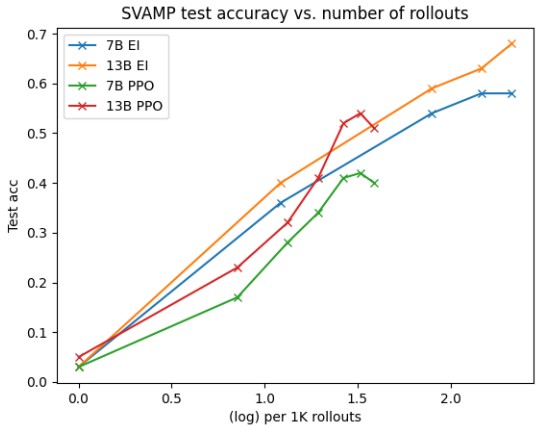

*Figure 5.* Sample complexities on SVAMP. Surprisingly, EI appears nearly as sample efficient as PPO.

### 4.3. Implementation Details

It is well known RL training can be quite sensitive to architectural and hyperparameter choices. This is even more so the case for LLM fine-tuning. In this section we ablate and discuss the factors we found most important in our tasks.

**PPO model architecture and training parameters** To save memory we use a joint architecture for the PPO policy and value heads. We found it important to use a relatively large value branch (L=4 transformer layers) and detach the gradients coming from the value branch to the policy trunk. Without detachment we found value gradients interfere with policy gradients, as similarly observed in Stiennon et al. (2020), causing instability with a big update to either branch. See Figure 17 which compares maj@1 score of a student with a large value branch and detached value gradients versus the default.

Low rank adaptation (LoRA) (Hu et al., 2021) with rank $r = 128$ helped significantly to further stabilize a full layer fine-tuning while still maintaining performance (Sun et al., 2023). A large enough batch size (BS = 256) and a small lr = 1e-6 also helped with stabilization. We additionally experimented with a partial fine-tune of only the top M layers. This saved memory but at the cost of a few percentage points of performance.

We also found a non-trivial KL penalty of $0.05$ to be critical for preventing model collapse after more than a hundred gradient updates. This is in contrast to Bai et al. (2022) who do not see a significant need for the KL constraint. We attribute its importance here to the somewhat unnatural distribution of text found in the the reasoning tasks which consist of broken natural language and computations enclosed in <<x+y=z>> tags. For tasks with distributions closer to pure natural language dialogue, such as those considered in Bai et al. (2022), the KL constraint seems less necessary.

**Sampling parameters affect exploration** We found the best temperature to use for good exploration during PPO training heavily depends on the initialization. When starting from an SFT checkpoint we choose T = 0.7. However, sampling on a high temperature when starting from the pretrained prompted model often results in collapse. In these cases we choose a low temperature (T = 0.2). Potentially better results for PPO could likely be achieved by annealing the exploration temperature over the course of training. We similarly experimented with the sampling temperature used during exploration in EI, ultimately deciding on $T = 1.0$ to maximize solution diversity without sampling too many degenerate solutions.

We also experimented with best K of N (KoN) sampling during PPO training to promote more solution diversity. In this setup the K highest reward samples of N rollouts from a single prompt are kept for training and the rest are discarded. Choosing parameters K ≪ N prioritize high reward samples and discard low reward ones, resulting in a training distribution more similar to the curated EI dataset.

However, one important consideration is the impact of the K/N ratio on training time and sample complexity, with smaller ratios taking proportionally longer. For example, K=1,N=8 takes 8 times as long as the default K=1,N=1. Further, we ultimately found little benefit to small K/N ratios with most configurations yielding decreased performance over K=1,N=1. In practice we found setting K=4, N=4 worked best. See Figure 6 which compares the performance of various choices of K and N.

**Model size and initialization affect exploration** We found both the quality of the student initialization and the size of the student significantly affected the type of exploration

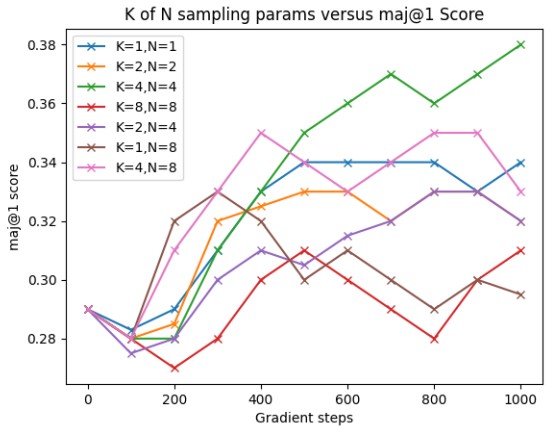

*Figure 6.* Best K of N sampling parameters versus maj@1 score during training. K=4, N=4 yields a fast runtime and best performance.

| | maj@1 | maj@96 | Rerank@96 | pass@96 |
|---|---|---|---|---|
| SFT[2] | 0.36 | 0.45 | 0.53 | 0.76 |
| SFT[4] | 0.41 | 0.47 | 0.54 | 0.72 |
| PPO[2] | 0.43 | 0.48 | 0.59 | 0.8 |
| PPO[4] | 0.44 | 0.49 | 0.58 | 0.77 |

*Table 4.* Results for full supervised fine-tune (SFT[4]), half supervised fine-tune (SFT[2]) and their PPO fine-tunes. Fine-tuning for only two epochs gets pass@96 = 0.76. This decreases to 0.72 with two additional epochs of fine-tuning.

engaged in during training. In particular **larger models engaged in more diverse exploration** while **models with worse generalization engaged in less diverse exploration** (See Appendix Section B). This in turn directly impacts model performance when trained on exploration data, with models engaging in more diverse exploration improving more from RL training.

To further examine the observations about overfitting, we supervise fine-tune a Llama 2-chat 7B model for half as many steps than the SFT model reported in Table 1. We call the former model SFT[4] and the latter SFT[2]. Despite half the training, SFT[2] has similar Rerank@96 and superior pass@96 scores to SFT[4] with the main difference being the maj@1 accuracies. When sampled K = 96 times on each train prompt, SFT[2] produces on average 3.7 unique correct solutions compared to SFT[4] which produces 2.9 unique correct solutions. We also find SFT[2] benefits significantly more from RL fine-tuning than SFT[4], jumping from maj@1=0.36 to maj@1=0.43. It's important to note some of this improvement also happens with continued SFT training, however at the cost to model output diversity and pass@96 performance.

We believe **RL fine-tuning is less prone to overfitting** when compared to static SFT fine-tuning precisely because of the exploration process which generates its own training data. This results in in more diverse solution paths than the SFT training set, ameliorating overfit. This is also in line with recent work that found RLHF to result in better (out-of-distribution) generalization than SFT on summarization and instruction following tasks (Kirk et al., 2023). This benefit can be found in both PPO and EI which have almost 10% pass@96 improvement over continued SFT (yet a much smaller pass@96 improvement over a light SFT). To support this hypothesis we plot the solution accuracies and diversities of EI models over each iteration in Figures 8 and 9, respectively. Figure 9 also shows larger models generate more diverse solutions.

## 5. Discussion and Broader Impact

As a result of our careful experimentation we uncovered several interesting findings involving the relative performance of different RL algorithms with SFT, their impact on various metrics, and the utility of dense/synthetically given rewards. These observations, taken together with the fast convergence of both online algorithms and the low-impact of ORM guidance and dense rewards, suggests models are not engaging in a significant amount of exploration beyond pre-training data. Crucial in our setting is the usage of a pretrained model imparting a strong exploration prior. Without such a prior, exploration in a high-dimensional textual action space would be very difficult. However, this prior also constrains the exploration engaged in at the beginning of training. Our results suggest this strong pre-training bias continues to significantly limit exploration throughout RL training, resulting in fast performance saturation. We view the discovery of new techniques encouraging complex, rich exploration of reasoning problems as fundamental to progress in LLM reasoning capability. More sophisticted prompting strategies such as Tree of Thought (Yao et al., 2023) and combining LLM generative abilities with evolutionary algorithms (Lehman et al., 2022) have already begun to make progress in this direction.

We also note prior work in RLHF finds PPO outperforms EI type approaches in human preference satisfaction and instruction following (Gulcehre et al., 2023; Dubois et al., 2023; Kirk et al., 2023). Importantly, in our setting we always have a reliable ground truth reward to optimize. However, in RLHF, models must optimize against an unreliable reward model, often resulting in over-optimization (Gao et al., 2022). The relatively superior performance of PPO over EI on RLHF tasks versus reasoning tasks suggests PPO better mitigates such over-optimization. This is perhaps not too surprising since PPO training penalizes student models diverging from the initial policy via both its clipped objective and additional KL-constraint.

**Broader Impact:** This paper presents work whose goal is to advance the field of large language modeling. As is the case with all works involving large language modeling, there are dangers involving the spread of misinformatioin and propagation of societal biases.

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

## A. RCRL Label Balance

We also experiment with different proportions of '[GOOD]' and '[BAD]' labels in RCRL training data. This is motivated by a desire to make better use of abundant negative data, which is much easier to generate than its positive counterpart. Better teaching the student what **not** to do with this data would ideally increase the number of valid solutions. Recall by default we balance the number of positive and negative samples.

We conduct experiments on LLama 2-chat 7B GSM8K without any SFT data. We apply only one round of Expert Iteration ($K = 1$ per question), producing a student model we refer to as **EI-minimal**. Note, in this setting we only provide '[GOOD]' and '[BAD]' labels for entire solutions, rather than providing labels at the step level. Results are reported in 5.

|  | positive:negative ratio | GSM8K (maj@1) |
|---|---|---|
| **EI-minimal** | - | 0.17 |
| **RCRL** | 100:1 | **0.18** |
|  | 10:1 | **0.18** |
|  | 1:1 | 0.15 |

*Table 5.* **RCRL** without SFT, using different proportions of positive and negative samples. As we increase the proportion of negative samples, performance generally decreases. At best, we only see very marginal gains using **RCRL**. Note: **EI-minimal** refers to running EI for one iteration, with $K = 1$ per question.

We find we achieve best performance when the amount of positive training data greatly outweighs the amount of negative data. In these cases, our RCRL models' maj@1 score slightly exceeds the maj@1 score of the data generating EI-minimal model. Yet, when we balance the amount of positive and negative training data, we find performance is degraded. This suggests our 7B student doesn't effectively learn from the provided negative demonstrations. We suspect either a larger model or an easier task would give better results.

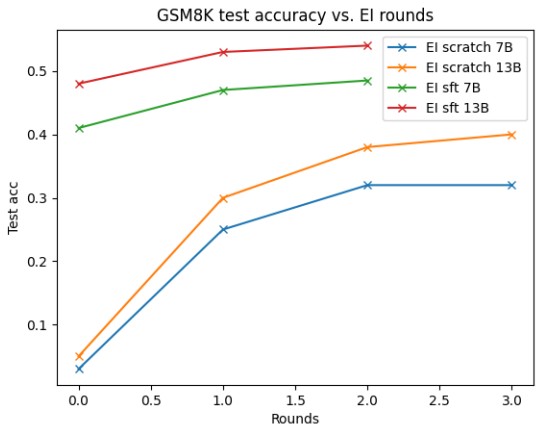

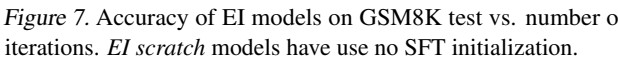

*Figure 7.* Accuracy of EI models on GSM8K test vs. number of iterations. *EI scratch* models have use no SFT initialization.

*Figure 8.* Accuracy of EI models on GSM8K test vs. number of iterations. $K = 4$ samples per prompt are used to construct a fine-tuning dataset for the next round.

## B. EI Improvement across Iterations

Figures 7 and 8 plot the maj@1 score of models on versus rounds of expert iteration. On both datasets the score is monotonically increasing until convergence after at most four rounds. Models initialized from an SFT checkpoint converge faster than their pretrained counterparts. Each round of expert iteration samples $K * \texttt{num\_train}$ rollouts, with the longest running training loop generate at most $5 * 4 * 7000 \approx 10^6$ samples.

Figure 9 reports the diversity of solutions across rounds of expert iteration as measured by two separate metrics for solution uniqueness. *exact diversity* checks for equality between two solutions using exact string match. *trace diversity* checks for equality between two solutions by first extracting the *trace* of a solution as the sequence of intermediate calculations used to get to the final answer. An exact match is then performed on this trace representation.

**Solution diversity increases then decreases over training** Overall both measures of solution diversity increase for both model sizes over the first two rounds of expert iteration. After the first two rounds both trace diversity appears to plateau and in some cases slightly decrease. Exact diversity continues to increase for 13B, but not at the same rate as during the first two rounds. The largest increases in solution diversity over the first two rounds also match when the largest gains in maj@1 performance occur. This lends evidence to the intuition that a high-performing student will be able to generate many correct but unique solutionst to the same problem. Further, we see during later rounds of expert iteration that while maj@1 score improves slightly, diversity suffers. This provides further evidence that training is begining to overfit to maj@1 score, in the process

reducing both pass@n and solution diversity. We see the same behavior

**Larger models generate more diverse solutions** The above figures also demonstrate the 13B model produces significantly more diverse outputs than the 7B model. This is true during every round of fine-tuning, with the gap getting larger as more training is done. Interestingly, the 13B model appears to produce an *exactly* unique solution with every sampling after 4 rounds of expert iteration. However, its trace diversity peaks after two rounds, indicating 13B tends to introduce semantic diversity without changing the underlying computational structure of a solution.

## C. Sample Complexities

In this section we plot all sample complexities on benchmarks accompanying the results in Section 4. Figures 11 and 12 report results on GSM8K without supervised fine-tuning. Figures 13 and 14 report results on SVAMP.

As in the SFT case, using an ORM to guide EI and PPO on prompted GSM8K models does somewhat reduce sample complexity but does not improve best performance (if anything the ORM reward slightly hurts converged maj@1 score). We see the same story when providing a dense ORM reward, further decreasing sample comlexity but at the cost of final converged performance. Our best results still come from using only the ground truth score. We suspect the performance degradation introduced by the ORM reward could be alleviated with a larger reward model. However, we do not believe using a larger model would improve over just the ground truth reward. Similar results are seen for SVAMP.

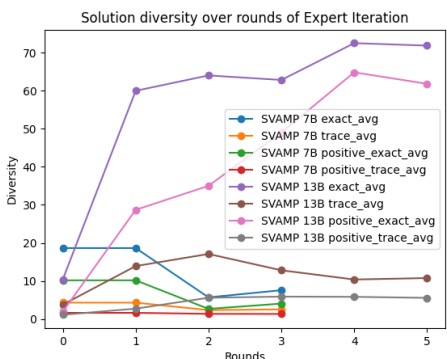
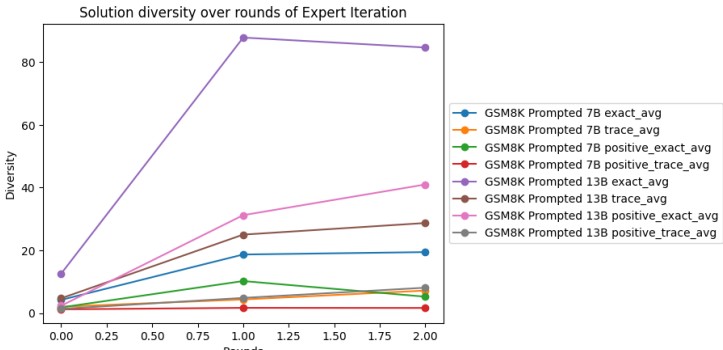

Figure 9. **Left:** Diversity of GSM8K model output over rounds of EI. (No SFT). **Right:** Diversity of SVAMP model output over rounds of EI. $K = 96$ samples are used per prompt. *positive* diversity measures diversity in the subset of solutions with a correct final answer.

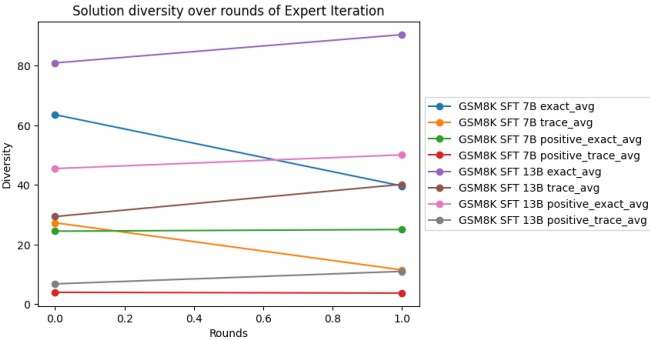

Figure 10. gsm8k sft diversity

## D. Curriculum Learning for RL

In addition to vanilla PPO we experiment with backtracking (Salimans & Chen, 2018) and Prioritized Level Replay (**PLR**) (Jiang et al., 2020) as algorithms from the curriculum learning literature. Such algorithms aim to construct a "curriculum" of subproblems, with the model ideally learning to generalize from easier subproblems to harder subproblems.

Backtracking in particular is a natural choice as it relies on using high-quality supervised trajectories to improve exploration of the solution space. This is done by sampling the student policy $\pi$ on the partially complete solution $(Q, P_i)$ where $P_i$ is a sequence of intermediate ground truth steps $(S_1, ..., S_i)$. The algorithm proceeds by setting an initial threshold $\tau_0 \in (0, 1)$ which represents how far back from the final answer to initialize partial solutions. By default we use $\tau_0 = 0.9$. Then, for each problem $Q$ which can be solved from $P_i$, we remove the last step $S_i$ and condition on $P_{i-1}$ the next time $Q$ is sampled.

PLR does not rely on access to SFT data, instead heuristically prioritizing problems with high "learning potential"

estimated by the average absolute advantage. Prioritizing problems with this potential allows the model to focus on problems that are neither too easy nor too hard, making efficient use of its exploration budget. We initialize the student using a supervised fine-tuned LLama 2-chat 7B on GSM8K. Results are reported in Figure 15.

Overall we find neither method exceeds the performance of default PPO. We hypothesize this is due to the limited exploration the model engages in from the start, due to both pre-training and supervised fine-tuning. We speculate better results might be achieved on a harder dataset with more intermediate steps, particularly when using backtracking.

## E. Data augmentation

We additionally experimented with generating synthetic $(Q, A)$ training pairs via an approach inspired by backtranslation (Sennrich et al., 2015). We assume access to a supervised fine-tuning dataset $\mathcal{D}$ of $(Q, A)$ pairs and train a $Q \rightarrow A$ model $M_{Q \rightarrow A}$ as our usual student model. We call this model the verifier. We can also utilize $\mathcal{D}$ to train models of the form $M_{A \rightarrow Q}$ and $M_{A \rightarrow A}$ which map answers to questions and answers to answers respectively. We train $M_{A \rightarrow Q}$ simply by fine-tuning the pretrained model $M$ to predict $p(A|Q)$ where $(Q, A) \sim \mathcal{D}$. We call the combination of $M_{A \rightarrow A}$ and $M_{A \rightarrow Q}$ the generator. We construct a train set for $M_{A \rightarrow A}$ as follows: For each $A$ in $(Q, A) \in \mathcal{D}$ we randomly sample three other answers $A_1, A_2, A_3$ from $\mathcal{D}$ which act as a conditional prompt. We then train $M_{A \rightarrow A}$ by minimizing $p(A|A_1, A_2, A_3)$.

We sample $M_{A \rightarrow A}$ on each ground truth answer $A \in \mathcal{D}$ $K = 8$ times, constructing a synthetic dataset of answers $\mathcal{A}$. We then use our backwards model $M_{A \rightarrow Q}$ to produce questions for each of the synthetic answers $A \in \mathcal{A}$. This forms a synthetic $(Q, A)$ dataset $\mathcal{D}_{\text{synth}}$. Finally, for each synthetic $(Q, A)$ pair, we sample our student model $M_{Q \rightarrow A}$

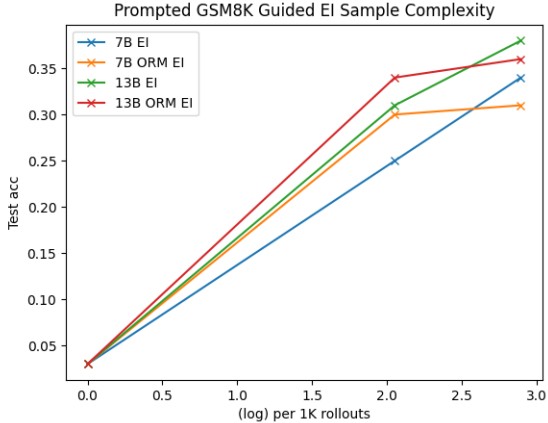

*Figure 11.* Sample complexity of default versus ORM guided EI students on GSM8K (no SFT). The ORM improves sample complexity initially but ultimately underperforms using only the ground truth.

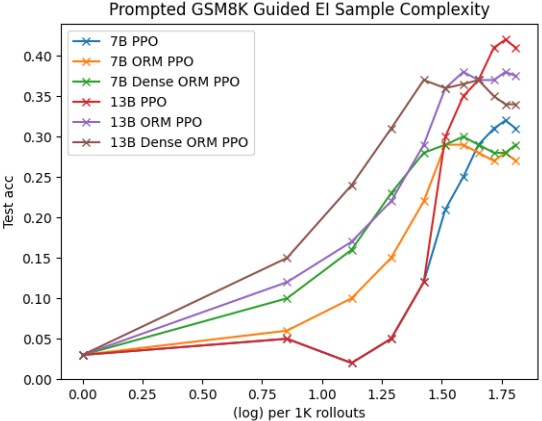

*Figure 12.* Sample complexity of default versus ORM guided PPO students on GSM8K (no SFT). Similarly to as in EI, the ORM improves maj@1 score over using only ground truth rewards but eventually underperforms.

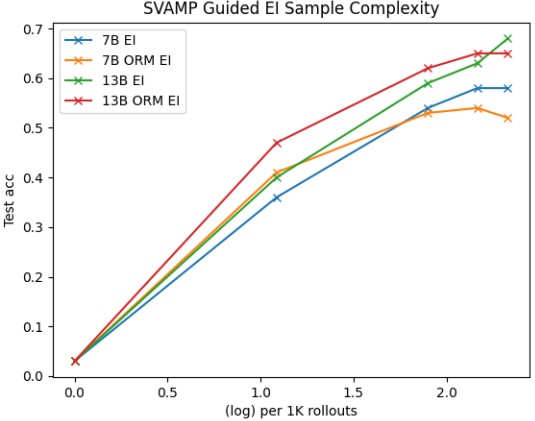

*Figure 13.* Sample complexity of default versus ORM guided EI students on SVAMP.

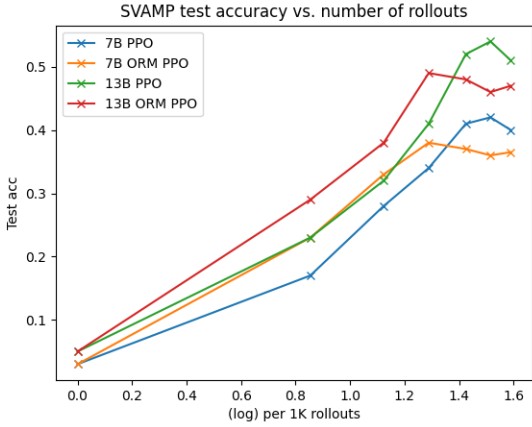

*Figure 14.* Sample complexity of default versus ORM guided PPO students on SVAMP.

$K = 20$ times for each question and check whether the student model's final answer agrees with the "intended" final answer. We refer to the percentage of student generated solutions recovering the intended final answer as the *score* for a synthetic $(Q, A)$ pair. We plot the distribution of scores in Figure 16.

We see that the majority of synthetic pairs, over 50,000, never have their solutions recovered by the student $M_{Q \to A}$. This is either because a) the student is too weak to solve the question or b) the question is impossible to solve. Either way, we likely do not want to include these new training data for the student. Similarly, we likely do not want to include questions which are always solved by the student, i.e. those with score = 1, as they are too easy. Additionally,

we should be wary of questions which have a small score in the range $(0, \epsilon)$. We expect many questions will have been solved incorrectly but still arriving at the correct final answer. We should exclude such problems from our training dataset.

We expect the highest quality data $(Q, A)$ to have a score in the neighborhood $(\frac{1}{2} - \tau, \frac{1}{2} + \tau)$. These questions should be not too hard but not too easy for our student. Figure 6 shows the performance of student models fine-tuned on a combination of ground truth data and synthetically generated data with scores in the range $(\frac{1}{2} - \tau, \frac{1}{2} + \tau)$. All models are trained for five epochs with an initial lr = 2e-5 cosine decayed to 2e-7. Llama 2-chat 7B is used as the pretrained base model.

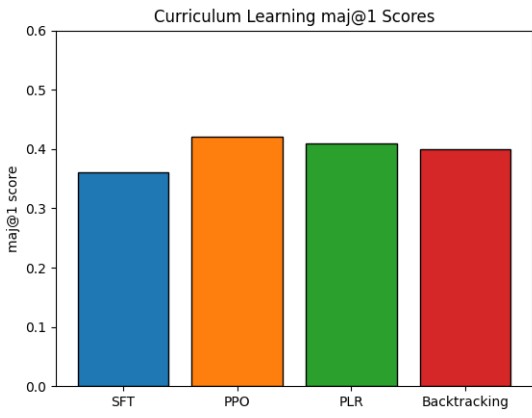

*Figure 15.* maj@1 scores on GSM8K for Prioritized Level Replay (PLR) and Backtracking techniques compared to default PPO and SFT.

|  | maj@1 |
| --- | --- |
| $\tau = 0.1$ | 0.38 |
| $\tau = 0.2$ | 0.36 |
| $\tau = 0.3$ | 0.34 |
| SFT | 0.41 |

*Table 6.* Performance of models training with various amounts of synthetic data vs. the SFT baseline. Note: $\tau$ represents the size of the neighborhood of scores around $\frac{1}{2}$ that are not filtered out.

Unfortunately, it seems introducing any amount of synthetically generated data degrades performance. When manually inspecting the synthetically generated $(Q, A)$ pairs it becomes clear why. There is an extremely high number of false positives. Consider the following example of a synthetic pair shown in Table E:

This is an example of a low-quality sample we do not want in our training data. Ideally, such a sample would have a score of 0 since the technically correct answer is 100, not 120. However, the SFT $M_{Q \to A}$ student we use to construct a score for each $(Q, A)$ sample computes the final answer as 120 a staggering 47% of the time. The verifier makes the exactly the same mistakes the $M_{A \to A}$ model made when constructing the question, likely because they were trained on similar distributions.

We suspect using a larger model more capable of detecting these sort of trivial non-solutions would do substantially better at generating backwards synthetic data. Similarly, employing separate models as the generator and verifier may reduce the probability of both making the same mistakes, improving the reliability of the score for each pair. We leave this as future work.

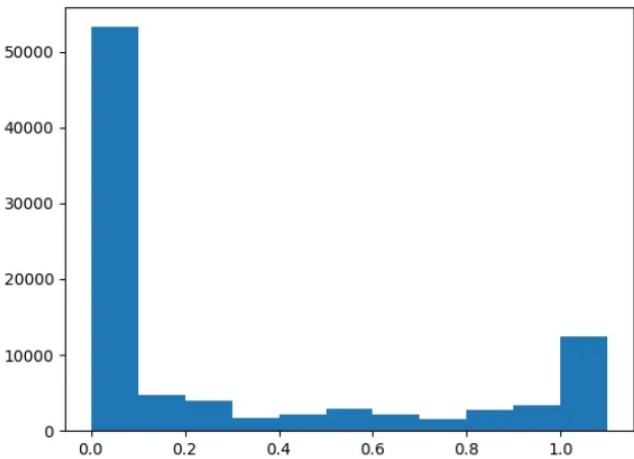

*Figure 16.* Scores of synthetically backwards generated $(Q, A)$ pairs. Note: the score refers to the percentage of times the forward student model $M_{Q \to A}$ recovers the intended final answer.

| Question | "A school of 100 musicians goes on a skiing trip. 40% are beginners, 30% are intermediate, and 50% are advanced. How many people went on the skiing trip?" |
| --- | --- |
| Answer | "There are 100 * 0.4 = 40 beginner skiers. There are 100 * 0.3 = 30 intermediate skiers. There are 100 * 0.5 = 50 advanced skiers. Therefore there are 40 + 30 + 50 = 120 skiers total." |

## F. RCRL Step-label Generating Process

Another natural candidate which could be used to identify mistakes at each step is a Process Based Reward Model (PRM) (Lightman et al., 2023). A PRM estimates the probability of correctness of a step $S_i$, $p(S_i \text{ correct}|Q, S_1, S_2, ..., S_i)$ independently of its impact on the final answer. However, this would be expensive, requiring collecting human annotated samples. Instead, we propose to approximate the *optimal value function* $V^*$ of the reasoning task. $V^*$ corresponds to the value function of the *optimal policy* which is able to successfully solve the reasoning task from any logically valid intermediate state $S_j$. Such an optimal value function would have $V^*(Q, S_1, ..., S_i) = 1$ for a solution prefix with no mistakes, and $V^*(Q, S_1, ..., S_i) = 0$ if the prefix already contains a mistake which will result in an incorrect final answer. Note however, $V^*$ does not exactly correspond to a PRM. This is because a partial solution $S_1, ..., S_i$ with a mistake at step $j \neq i$ and valid terminal step $S_i$ will have $V^*(Q, S_1, ..., S_i) = 0$ and $PRM(Q, S_1, ..., S_i) = 1$. To make this distinction clear, we call models we train to

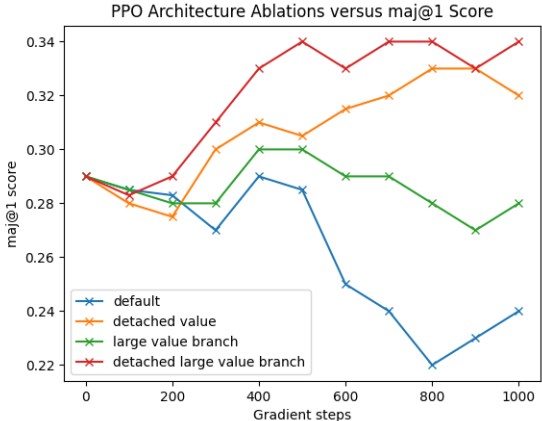

*Figure 17.* Comparison of reward curves for PPO architecture ablations. Using both gradient stopping and a larger value head works best.

directly approximate $V^*$ stepwise ORMs or **SORMs**.

## G. PPO Architecture Ablations

## H. CommonsenseQA Benchmark

In addition to the mathematical reasoning tested in GSM8K and SVAMP, we also conduct evaluations on the CommonsenseQA (CSQA) (Talmor et al., 2018) benchmark. CSQA is a dataset of 12,247 commonsense reasoning questions, testing reasoning that requires leveraging some amount of prior world knowledge. Each question has 5 possible multiple choice options, with a single correct answer. We use a sparse reward based on whether or not the model generation for a particular question results in the correct choice.

|  | maj@1 |
|---|---|
| Few-shot prompting | 0.63 |
| SFT | 0.70 |
| $\text{EI}_n$ | 0.76 |
| RCRL | 0.72 |
| PPO | **0.77** |

*Table 7.* Performance on the CSQA evaluation set with the Llama2 7B model. Few-shot prompting uses the same 7 examples as in (Zelikman et al., 2022) with no model finetuning. For $\text{EI}_n$, RCRL, and PPO we initialize with SFT and report the eval accuracy at convergence.

Similarly to the math-based reasoning benchmarks in Section 4, we also find that $\text{EI}_n$ and PPO perform on par for CSQA, while RCRL underperforms both methods. This suggests that our findings could extend beyond math-based reasoning—where EI and PPO are similarly effective, while RCRL models seem unable to learn from self-generated negative examples.

