# OpenReview forum: "Teaching Large Language Models to Reason with Reinforcement Learning"
_ICML.cc/2024/Workshop/AI4MATH — ICML 2024 Workshop AI4MATH Poster_

### Official Review · Reviewer_6v9v · 2024-06-12

**Rating:** 8
**Confidence:** 3

**Summary:**

This paper examines multiple reinforcement learning (RL) algorithms, including Expert Iteration, Proximal Policy Optimization (PPO), and Return-Conditioned RL, focusing on enhancing the reasoning capabilities of large language models (LLMs).

Specifically, the study:
1. Conducts an investigation of PPO fine-tuning with varied rewards, model sizes, and initializations.
2. Introduces a comparison between Expert Iteration and Return-Conditioned RL.
3. Integrates Outcome Based Reward Modeling (ORM) with the aforementioned RL algorithms.

The findings demonstrate that Expert Iteration consistently outperforms others in terms of effectiveness, while exhibiting a surprisingly similar sample complexity to PPO. The paper also elaborates on the implementation specifics to address the sensitivity concerns associated with RL training and discusses the future of these approaches, identifying exploration as a significant limiting factor.

**Questions:**

Please refer to Reasons to Reject.

**Reasons To Accept:**

- Originality: The paper presents a comprehensive comparison of mainstream RL algorithms applied to the fine-tuning of LLMs.
- Quality: The study defines a clear research scope, including the framework, dataset, and evaluation settings, and conducts extensive experiments to compare the algorithms. The results are robust and provide valuable insights.
- Clarity: The paper clearly articulates the research problem, algorithm details, and experimental settings.
- Significance: The findings are likely to benefit the community, particularly in reasoning-related tasks with LLMs (e.g., mathematics, coding) where RL is increasingly being applied.

**Reasons To Reject:**

- The datasets used in the experiments, namely GSM8K and SVAMP, are relatively simple and limited in scope. Given the rapid advancements and near-resolution of challenges within these datasets, the community's focus may shift towards more challenging datasets such as MATH. It would be beneficial to see the limits of RL on these more complex problems to better understand its potential impact.

- It would be highly beneficial for the community if the code and data were made available for reproduction purposes. Releasing these resources would greatly enhance the paper's impact by facilitating further research and validation of the results.
- The SFT model appears less competitive in terms of performance. Have you considered replacing it with state-of-the-art models such as MetaMath[1] or Deepseek-Math-7b-RL[2] on GSM8K? It would be insightful to determine whether the findings still hold with these advanced models.

[1] Yu, Longhui, et al. "Metamath: Bootstrap your own mathematical questions for large language models." arXiv preprint arXiv:2309.12284 (2023).

[2] Shao, Zhihong, et al. "Deepseekmath: Pushing the limits of mathematical reasoning in open language models." arXiv preprint arXiv:2402.03300 (2024).

---

### Official Review · Reviewer_rT3L · 2024-06-12

**Rating:** 7
**Confidence:** 3

**Summary:**

The paper provides a study comparing various RL algorithms for fine-tuning LLMs on reasoning tasks. The primary focus is on Expert Iteration (EI), Proximal Policy Optimization (PPO), and Return-Conditioned RL (RCRL), with EI demonstrating superior performance and sample efficiency across multiple metrics. The study emphasizes the significant role of supervised fine-tuning (SFT) in enhancing model performance and highlights the challenges posed by the pretrained models' bias, which limits exploration during training. The authors also note that larger models and more diverse training data improve performance, underscoring the need for new techniques to encourage richer exploration. The findings suggest that overcoming these challenges is essential for advancing LLM reasoning capabilities.

**Questions:**

How do you anticipate the proposed methods will perform on non-mathematical reasoning tasks, and have you considered testing on a broader range of benchmarks to validate the generalizability of your approach?

**Reasons To Accept:**

The paper is well-motivated and presents a timely exploration of applying RL techniques to enhance LLM reasoning capabilities, a natural progression from previous RL successes in other domains. The work is well supported by a comprehensive comparison of multiple RL algorithms, demonstrating that Expert Iteration consistently achieves superior performance with competitive sample efficiency. Furthermore, the study's insightful observations about the impact of supervised fine-tuning, sample complexity, and exploration strategies provide valuable guidance for future research. The paper is well-written and easy to follow.

**Reasons To Reject:**

Overall very good paper.

Minor comments:
- Include more detailed ablation studies to better understand the contributions of each component in the proposed methods.
- Incorporating more diverse benchmarks beyond math problems could also strengthen the generalizability of the findings.

---

### Official Review · Reviewer_iY2u · 2024-06-13

**Rating:** 5
**Confidence:** 3

**Summary:**

This paper explores the application of various RL algorithms to enhance the reasoning capabilities of LLMs. The authors investigate and compare several RL algorithms, including Expert Iteration, PPO, and ORM-based methods, in terms of performance and sample efficiency. Experiments are conducted on the GSM8K and SVAMP datasets. Through these experiments, Expert Iteration is found to have better generalization and superior training efficiency compared to PPO, while ORM does not provide any significant improvements.

**Questions:**

1. I would like to know how you trained the ORM. Did you create a separate RM dataset (including some never-before-seen questions for the policy model to generate positive and negative examples) to train the ORM, or did you directly use the questions seen during the SFT stage to generate positive and negative examples?

**Reasons To Accept:**

1. Researching how to use RL algorithms to guide finetuning is highly significant.
2. The authors conducted detailed experiments on GSM8K and SVAMP, yielding some interesting findings.

**Reasons To Reject:**

1. Insufficient Experiments: Although the author conducted extensive experiments on GSM8K and SVMAP, these comparisons were not extended to the more challenging MATH dataset. Conducting experiments on the MATH dataset is crucial. The pretrained model may already perform well on GSM8K due to potential data leakage, leading to randomness in the experimental results. It might not be that the EI method is inherently more useful to improve reasoning abilities compared to PPO, but rather that the EI method is more suited to simpler datasets. To thoroughly validate the claims of this paper, it is essential to perform experiments on the MATH dataset.

2. Unsatisfactory paper writing: a) There are some typos in paper, such as "in in" at line 387. b) The formula in line 112. c) The Figure presentation is not intuitive enough. For instance, in Figure 1, the EI method plots 3 points, while the PPO method plots 9 points. What does this mean? Please clarify, as this way of plotting makes the comparison seem unfair.

3. Some relevant papers[1,2] are missing. For instance, in [1], they discovered that ORM and even some more dense PRM models can enhance performance on the MATH dataset, improving generalization from easy to hard problems. This needs to be cited or discussed.

[1] Easy-to-hard generalization: Scalable alignment beyond human supervision. 	arXiv:2403.09472

[2] Beyond human data: Scaling self-training for problem-solving with language models. arXiv:2312.06585

---

### Meta-Review · Area_Chair_bv6x · 2024-06-13

**Recommendation:** Accept (Oral, top 2)
**Confidence:** 4

**Metareview:**

This paper studies various RLHF approaches for enhancing the reasoning capabilities of LLMs. The paper not only presents a comprehensive comparison of multiple RL algorithms, but also provides insightful observations about the impact of supervised fine-tuning, sample complexity, and exploration strategies. The findings are likely to benefit the community. The authors could further improve the paper by conducting studies on a more challenging dataset such as MATH.

---

### Decision · Program_Chairs · 2024-06-13

Accept (Poster)